# The Effect of Cardiopulmonary Resuscitation (CPR) Education on the CPR Knowledge, Attitudes, Self-Efficacy, and Confidence in Performing CPR among Elementary School Students in Korea

**DOI:** 10.3390/healthcare11142047

**Published:** 2023-07-17

**Authors:** Jang-Sik Ko, Seon-Rye Kim, Byung-Jun Cho

**Affiliations:** 1Department of Paramedicine, College of Health Science, Kangwon National University, 346 Hwangjo-gil, Dogye-up, Samcheok-si 25945, Republic of Korea; diver114kr@naver.com; 2Department of Healthcare Management, College of Health Science, Youngsan University, 288 Junam-ro, Yangsan-si 50510, Republic of Korea

**Keywords:** cardiopulmonary resuscitation education, confidence, elementary school students, knowledge, attitudes, self-efficacy

## Abstract

Cardiopulmonary resuscitation (CPR) education for schoolchildren is emphasized, as bystander CPR is a vital key to increasing the survival rate of out-of-hospital cardiac arrest (OHCA) victims. This study was conducted to verify the effect of CPR education on knowledge, attitudes, self-efficacy, and confidence of Korean elementary school students in performing CPR. Data were collected through structured questionnaires before and after CPR education and analyzed using descriptive statistics, *T*-tests, and hierarchical regression. Significant improvements in CPR knowledge, attitudes, self-efficacy, and confidence in performing CPR were found after CPR education, with the greatest increase observed in confidence (*p* = 0.000). The influencing factors on confidence in performing CPR were school grade, attitude, and self-efficacy. Although a significant increase in schoolchildren’s CPR knowledge after education was shown, knowledge did not affect confidence in performing CPR. Therefore, early CPR education which focuses on improving confidence in performing CPR is recommended. CPR education might raise attitude and self-efficacy leading to increased confidence in performing bystander CPR. In conclusion, early and regular CPR education for elementary school students is crucial and should be conducted repeatedly.

## 1. Introduction

Recently, there has been a significant increase in the number of cardiac arrests (CA) worldwide, making it a major public health issue [1,2]. Cardiopulmonary resuscitation (CPR) plays a key role in circulating blood and delaying brain damage [3,4]. The survival rate of CA was more than three times higher when bystander CPR was conducted compared to individuals who did not receive CPR [2,5].

The significance of early CPR and CPR education for the public has been emphasized to enhance the survival rate of out-of-hospital arrest (OHCA) patients [2,6]. Bystander CPR is vital for enhancing the survival rate and neurological outcomes of OHCA victims [7]. Previous studies have indicated that at least 15% of the population should be educated in CPR in order to significantly increase the survival rate of OHCA. However, this goal is difficult to accomplish through voluntary education for the public. Thus, mandatory education for schoolchildren might be important [8,9]. Early CPR education for schoolchildren should be increased [10] as it could lead to long-term changes [9].

CPR education in schools has been supported by the World Health Organization (WHO) [8]. In Korea, CPR education for schoolchildren has enhanced the bystander CPR rate, resulting in a higher survival rate for OHCA victims and a decline in health care costs [11]. As schoolchildren could be more motivated and quicker to learn [12,13], CPR education should begin early. The American Heart Association (AHA) has recommended including 2 h of CPR education per year in school curricula [14]. In Korea, CPR education for schoolchildren is already mandatory [15]. However, the proper age to begin CPR education has remained controversial. Schoolchildren could remember CPR knowledge and skills, and they might pass them on to their family and friends [16]. The benefits of early CPR education were reflected in attitude, self-efficacy, confidence in performing CPR, intention to help others, and improved empathy [17,18,19,20,21].

Behavioral change might be more effective when supported by proper education, especially in public health issues [22]. A previous study on the intention-focused model of bystander CPR reported that a change in intention leading to behavioral changes could result in improved CPR outcomes [23]. Ajzen et al. claimed that intentions are estimators of motivation to conduct a behavior [24]. Based on previous studies showing that CPR education promoted willingness, confidence in performing CPR, and motivation to help others [22,23,24], we expected CPR education to influence CPR knowledge, attitudes toward CPR, self-efficacy for CPR, and confidence in performing CPR. We believed that schoolchildren could learn the value of life through learning CPR. The purpose of this study was not only to teach CPR skills but also to educate schoolchildren about the value of life. Various studies have been carried out on CPR education in elementary school students, but few studies have attempted to identify the factors influencing confidence in performing bystander CPR. Therefore, this study aimed to improve CPR knowledge, attitude toward CPR, self-efficacy for CPR, and confidence, and to emphasize the importance of CPR education. We examined the effect of CPR education on CPR knowledge, attitude toward CPR, self-efficacy for CPR, and confidence in performing CPR among elementary school students. Furthermore, we explored the influencing factors on confidence in performing CPR.

## 2. Materials and Methods

### 2.1. Study Design and Subjects

In this study, we conducted a Pre-Post Design including CPR education. This study was conducted in May 2021 and included 140 schoolchildren in the third, fourth, fifth, and sixth grades of an elementary school. A month before the start of the study, we visited an elementary school in rural areas. After obtaining permission from the principal of the elementary school, the authors visited all classrooms in the 3rd to 6th grades, respectively. We explained the purpose and process of this study and distributed two informed consents to students who showed interest in the study and expressed their intention to participate. Students who completed the submission of informed consent for themselves and their parents became the final subjects of this study. Informed consent was received from all participants and their parents before participating in the study. The study was approved by the ethics committees of Kangwon National University (reference number KWNUIRB-2020-11-005-003). This study was conducted in three steps. In the first step, the participants completed self-report questionnaires regarding general characteristics, CPR knowledge, attitude toward CPR, self-efficacy for CPR, and confidence in performing CPR. The structured questionnaire was written in Korean. In the second step, the students participated in two hours of CPR education by certified instructors. For the objective of this study, instructors certified by AHA used the standardized CPR program provided by the Korean Association of Cardiopulmonary Resuscitation [2]. The practical CPR education was performed on a Brayden Pro (Innosonian^®^) according to the AHA guidelines [25]. In the third step, the participants completed the same questionnaires used in the first step.

### 2.2. General Characteristics

The general characteristics of participants consisted of gender, school grade, number of CPR education, and the elapsed period after the last CPR education. School grade was categorized into four groups: third grade, fourth grade, fifth grade, and sixth grade of elementary school. The number of CPR education was classified into five groups: 0, 1, 2, 3, and ≥4. The elapsed period after the last CPR education was classified into four categories: not available, under 6 months, between 6–9 months, and over 9 months.

### 2.3. CPR Knowledge

The CPR knowledge referred to the basis of theoretical knowledge that should be accompanied by CPR, and the questionnaires used in the study of Patsaki et al. [17] were modified based on the basic CPR guidelines presented by the AHA in 2015 [12]. The questionnaires were verified on validity by two emergency rescue professors. The questionnaires consisted of eight questions about measuring CPR knowledge for the public [17,18,19], including CPR process, consciousness confirmation, chest compression, airway maintenance, artificial respiration, circulation confirmation, and brain damage. Each question evaluated 1 point for the correct answer and 0 points for the wrong answer. The score range for knowledge was 0 to 8, and a higher score represented better knowledge. In this study, the tool reliability was Cronbach’s α = 0.79.

### 2.4. Confidence in Performing CPR

Confidence referred to confidence in one’s ability to perform CPR, and the questionnaire was developed by Park [26]. In this study, confidence consisted of four questions, and the answer to each question was evaluated as “yes“ (1 point) and “no” (0 points). The range of scores was from 0 to 4, and a higher score represented stronger confidence in performing CPR. The reliability in this study was Cronbach’s α = 0.79.

### 2.5. Attitude toward CPR

Attitude toward CPR referred to the perception and willingness to implement CPR, and the questionnaire was developed by Kim et al. [25] and modified and supplemented to suit younger students. The tool was validated by two experts in the Department of Emergency Rescue. It consisted of 11 questions, and each question was scored with 5 points for “strongly agree”, 4 points for “agree”, 3 points for “average”, 2 points for “disagree”, and 1 point for “strongly disagree”. Each negatively worded question was calculated by reversing the score. The reliability of this study was Cronbach’s α = 0.70.

### 2.6. Self-Efficacy for CPR

Self-efficacy was a judgment on an individual’s ability to perform CPR. To effectively measure self-efficacy for CPR, a tool modified by Schlessel et al. [27] consisted of 17 questions, and each question was scored with 5 points for “strongly agree”, 4 points for “agree”, 3 points for “average”, 2 points for “disagree”, and 1 point for “strongly disagree”. Each negatively worded question was calculated by reversing the score. The reliability in this study was Cronbach’s α = 0.92.

### 2.7. Statistical Analysis

Using IBM software SPSS v.25.0, descriptive statistics were used to present general characteristics and each question of confidence. Independent *T*-test was used to analyze the differences in CPR knowledge, attitude toward CPR, self-efficacy for CPR, and confidence in performing CPR before and after CPR education. The chi-square test was used to analyze the difference of “yes” answers in each question of confidence. The influence on confidence in performing CPR was analyzed using hierarchical regression.

## 3. Results

### 3.1. Baseline Characteristics of Study Participants

Of the students, 72 (51.4%) were male. The participants were distributed similarly across the 3rd to 6th grades, and the majority of them had received CPR education at least once, with only 5.7% having no prior experience. The number of CPR education instances was 37.1% for one time, followed by 25.7% for two times. Regarding the period since their last education, 40.7% had elapsed between 6 to 9 months (Table 1).

### 3.2. CPR Knowledge, Attitudes toward CPR, Self-Efficacy for CPR, and Confidence in Performing CPR before and after CPR Education

The CPR knowledge before and after CPR education was 4.55 ± 1.49, and 5.01 ± 1.35, respectively. The difference in CPR knowledge before and after CPR education was significant (*p* = 0.008). The attitude toward CPR before and after CPR education was 3.48 ± 0.61, and 3.66 ± 0.59, respectively. The difference in the attitude toward CPR before and after CPR education was significant (*p* = 0.017). The self-efficacy for CPR before and after CPR education was 3.22 ± 0.75, and 3.43 ± 0.65, respectively. The difference in the self-efficacy for CPR before and after CPR education was significant (*p* = 0.013). The confidence in performing CPR before and after CPR education was 2.03 ± 1.20, 3.05 ± 0.98, respectively. The difference in confidence in performing CPR before and after CPR education was significant (*p* = 0.000). After CPR education, significant improvements were observed in all variables, with the greatest enhancement seen in confidence (Table 2).

### 3.3. Confidence Questions before and after CPR Education

The results showed significant improvements in all questions regarding the participants’ confidence in performing CPR (Table 3). The question that showed the most improvement was as follows: “When you find a cardiac arrest patient in front of you, are you confident in performing CPR?”. The number of schoolchildren who answered “Yes” increased from 51 (36.4%) to 112 (80.0%). Similarly, the question, “Can you use an AED if you find a cardiac arrest patient in the absence of medical staff?” improved from 48 (34.3%) to 78 (55.7%) in the number of schoolchildren who marked “Yes”. The question, “Can you perform CPR if you find a cardiac arrest patient in the absence of medical staff?” improved from 83 (59.3%) to 105 (75.0%) in the number of schoolchildren who marked “Yes”. The question, “Can you perform CPR if your family member has a cardiac arrest?” ranked highest both before and after CPR education, with 102 (72.9%) and 129 (92.1%) respondents marking “Yes”, respectively. There are the results of CPR knowledge questions, attitude questions, and efficacy questions before and after CPR education in Appendix A.

### 3.4. Hierarchical Regression of Influencing Factors on Confidence in Performing CPR before CPR Education

Before CPR education, the factors influencing confidence were school grade (β = 0.249), number of educational experiences (β = 0.191), attitude toward CPR (β = 0.214), and self-efficacy for CPR (β = 0.363), with an explanatory power of 47.3% (Table 4). This means that higher grade, more educational experiences, a more positive attitude toward CPR, and higher self-efficacy for CPR were associated with higher confidence in performing CPR.

### 3.5. Hierarchical Regression of Influencing Factors on Confidence in Performing CPR after CPR Education

After CPR education, the factors influencing confidence were school grade (β = 0.213), attitude toward CPR (β = 0.192), and self-efficacy for CPR (β = 0.215), with an explanatory power of 18.0% (Table 4). This indicates that higher grade, a more positive attitude toward CPR, and higher self-efficacy for CPR were associated with higher confidence. The number of CPR education was not a significant influencing factor immediately after CPR education (Table 5).

## 4. Discussion

Early bystander CPR plays a crucial role in saving OHCA patients and promoting the survival rate. Nevertheless, the worldwide rate of bystander CPR is less than 50% on average [4]. In our study, we investigated the effect of CPR education on CPR knowledge, attitudes toward CPR, self-efficacy for CPR, and confidence in performing CPR among elementary school students. This study reported significantly improved knowledge, attitudes, self-efficacy, and confidence in performing CPR after CPR education. Additionally, this finding indicated that CPR education could increase not only CPR knowledge but also confidence in performing CPR among schoolchildren. The enhanced confidence through CPR education should ultimately increase the rate of bystander CPR.

School-based CPR education has been advocated by the AHA and the WHO [28,29]. CPR education is the most efficient in the upper grades of elementary school [25,30,31], and their education has significant ripple effects on the people around them. Since 2009, as health education for fifth and sixth graders in elementary school has become compulsory discretionary in Korea [15], opportunities for CPR education have increased, and 82.9% of fifth and sixth graders in elementary school have CPR education experience [25]. Since the experience of CPR education and the willingness to perform bystander CPR are closely related [32], systematic CPR education should be conducted from an early age.

In our study, students showed improved knowledge after CPR education. The average knowledge score increased from 4.55 before CPR education to 5.01 (out of a total score of 8). Previous research has also reported greater knowledge after CPR education [25,26,27,30,31,32]. Therefore, early introduction of CPR education is needed [20,33,34,35]. Younger students could have a bigger capacity to learn CPR compared to older students [35]. One study even reported that the ability to perform CPR in 10-year-old students was similar to that of 13-year-old students in Germany [36]. Additionally, students between the ages of 10 and 11 had enough intelligence and appropriate weight to perform effective chest compression [19,20]. However, it is important to note that we investigated a relatively short interval of two hours between the CPR education and the second testing procedure. This timeframe might make the study resemble more of a training program rather than a comprehensive scientific investigation of the long-term effects of CPR education. A future study with longer follow-up periods is needed.

In addition, we compared attitudes toward CPR, self-efficacy for CPR, and confidence in performing CPR before and after CPR education. After CPR education, students significantly increased their attitude, self-efficacy, and confidence. The improved CPR knowledge might lead to improved attitudes, self-efficacy, and confidence in performing CPR. Especially, this study showed that students had greater confidence in performing CPR after CPR education. Several research studies have also proposed similar conclusions, indicating that CPR education boosted the confidence of schoolchildren [16,17,34]. Furthermore, CPR education for students significantly enhanced their self-worth and moral responsibility toward others [34]. Additionally, CPR education decreased the fear of making mistakes, raised confidence, and promoted a positive attitude in schoolchildren towards helping others [19,20,23,31,32]. Our results also revealed a significant increase in confidence in performing CPR on both strangers and family members after CPR education. Prior to education, only 36.4% of students were confident in performing CPR on strangers. However, after education, 80.0% of students expressed confidence, which was higher than the reported rate in Germany (72.3%) [36]. This improvement could be attributed to the effect of CPR education. A prior study reported that CPR education experience and CPR knowledge did not directly influence the willingness to perform bystander CPR [37]. Therefore, it is crucial to focus on improving confidence in performing bystander CPR during CPR education [37].

Our findings, which indicated that elementary school students’ attitudes and confidence in performing CPR increase with higher grades, were consistent with the research of Lubrano et al., who verified the effects of CPR education for children aged 8 to 11 years [38]. In this study, the attitude toward CPR was measured as an item that included confidence in performing CPR, suggesting that a more positive attitude correlates with higher confidence. Our study was consistent with previous studies that have reported significant increase in attitude, self-efficacy, and confidence in performing CPR after educating elementary school students [39,40]. Earlier education is highlighted as important because CPR education can increase the willingness, attitude, and intention to help others in young students [17]. Elementary school students are more easily motivated, and educational continuity and allocation of educational time can be maintained compared to adults, leading to more effective education [39]. Furthermore, several studies have reported that early CPR education helps young students learn the importance of helping others [41]. Additionally, Kim et al. demonstrated that CPR education for elementary school students is more effective than for the public [42]. The AHA and WHO recommend mandatory CPR education for schoolchildren as it would strongly influence the survival rate of OHCA victims [8,14]. In Korea, a health curriculum was introduced for 5th and 6th graders using discretionary activity time in elementary schools according to the notice by the Ministry of Education, Science and Technology in 2009 [15]. To increase the rate of bystander CPR in emergencies, systematic and repetitive CPR education for lower-grade elementary school students might be essential.

Previous studies have shown a positive correlation between CPR knowledge, positive attitude, and stronger confidence in performing CPR [42,43,44]. In addition, more education and higher educational satisfaction contributed to stronger confidence in performing CPR. Thus, regular, and repeated CPR education should be provided. In addition, various programs should be developed to increase educational satisfaction [45,46]. In one study, a first attempt to expand the use of CPR in school, a police officer even reported that high school students who had been more likely to participate in CPR courses, were more responsive and receptive than police officers who were often the first to intervene. The result that high school students’ confidence and early access to CPR were excellent compared to police officers strongly supported the need to regularize schoolchildren’s CPR education to improve bystander CPR. [47]. However, further research on the timing of retraining is necessary, as there are numerous studies on the necessary retraining period to increase performance confidence through periodic and repetitive education. In multiple regression analysis, the factors that influenced confidence in performing CPR among elementary school students were school grade, attitude, and self-efficacy, but not CPR knowledge. Therefore, systematic, and repetitive CPR education must be provided for elementary school students to increase their confidence in performing CPR and promote the provision of bystander CPR during emergencies.

In summary, CPR education significantly enhanced knowledge, attitude, self-efficacy, and confidence in elementary school students. These results highlighted the positive effects of CPR education for schoolchildren, emphasizing the need for early CPR education. The factors influencing confidence in performing CPR were school grade, attitude, and self-efficacy both before and after CPR education. Our finding provided a foundation for policy-makers to emphasize regular and repeated CPR education focused on improving confidence in performing CPR for elementary school students. In conclusion, early CPR education for elementary school students is crucial and should be conducted regularly and repeatedly as part of the school curriculum.

There are some limitations to this study. First, we only measured schoolchildren’s knowledge, and measuring CPR skills could lead to different conclusions. Therefore, further studies focusing on CPR skills among schoolchildren after CPR education are suggested. Second, this study used a non-standardized questionnaire that was not psychometrically tested for sensitivity. Third, the sample size was relatively small. Fourth, as the questionnaires were anonymous, we used independent *T*-tests. This limited our ability to measure the change in each student individually, which is a limitation of this study. Therefore, we attempted to demonstrate the effectiveness of CPR education by assessing the mean differences of each variable before and after CPR education. These limitations should be considered as the basis for our results.

## 5. Conclusions

The results of this study clearly demonstrated the positive effects of CPR education on young schoolchildren. We found that school grades, attitudes, and self-efficacy were important factors influencing confidence in performing CPR. Therefore, regular, and repeated CPR education focusing on improving confidence in performing CPR should be provided for elementary school students in order to increase the rate of bystander CPR for OHCA victims.

## Figures and Tables

**Table 1 healthcare-11-02047-t001:** Baseline characteristics of study participants.

Characteristics	*n*	%
Gender		
Male	72	51.4
Female	68	48.6
School Grade		
3rd grade	35	25.0
4th grade	33	23.6
5th grade	35	25.0
6th grade	37	26.4
Number of CPR education		
0	8	5.7
1	52	37.1
2	36	25.7
3	15	10.7
≥4	29	20.7
The elapsed period after the last CPR education		
Not available	8	5.7
<6 months	39	27.8
6–9 months	57	40.7
>9 months	36	25.7

**Table 2 healthcare-11-02047-t002:** A comparison of CPR knowledge, attitude toward CPR, self-efficacy for CPR, and confidence in performing CPR before and after CPR education.

Variables	Scale Range	Mini; Max	Before CPR Education	After CPR Education	*t*-Value	*p*-Value
CPR knowledge	0–8	1; 8	4.55 ± 1.49	5.01 ± 1.35	2.684	0.008
Attitude toward CPR	1–5	1.73; 4.91	3.48 ± 0.61	3.66 ± 0.59	2.410	0.017
Self-efficacy for CPR	1–5	1.71; 5.0	3.22 ± 0.75	3.43 ± 0.65	2.504	0.013
Confidence in performing CPR	0–4	0; 4	2.03 ± 1.20	3.05 ± 0.98	7.790	0.000

**Table 3 healthcare-11-02047-t003:** Confidence questions before and after CPR education.

Questions	Before CPR Education *n* (%)	After CPR Education *n* (%)	Pearson Chi-Square	*p*-Value
Can you perform CPR, if you find a cardiac arrest patient in the absence of medical staff?	83 (59.3%)	105 (75.0%)	7.835	0.005
Can you perform CPR, if your family member has cardiac arrest?	102 (72.9%)	129 (92.1%)	18.033	0.000
Are you confident in performing CPR, when you find a cardiac arrest patient in front of you?	51 (36.4%)	112 (80.0%)	54.632	0.000
Can you use an AED, if you find a cardiac arrest patient in the absence of medical staff?	48 (34.3%)	78 (55.7%)	12.987	0.000

**Table 4 healthcare-11-02047-t004:** Hierarchical regression of influencing factors on confidence in performing CPR before CPR education.

Variables	Model 1				Model 2				Model 3			
	B	SE	β	*t*	B	SE	β	*t*	B	SE	β	*t*
Constant	0.062	0.808		0.077	−1.344	0.508		−2.644 **	−2.963	0.773		−3.831 ***
Gender	−0.052	0.185	−0.022	−0.279					0.009	0.159	0.004	0.057
School Grade	0.266	0.111	0.252	2.407 *					0.263	0.092	0.249	2.860 **
Number of CPR education	0.271	0.102	0.264	2.648 **					0.196	0.088	0.191	2.235 *
The elapsed period after the last CPR education	0.000	0.128	0.000	−0.002					−0.007	0.109	−0.005	−0.065
CPR knowledge					0.036	0.060	0.044	0.597	−0.025	0.056	−0.032	−0.452
Attitude toward CPR					0.194	0.173	0.099	1.118	0.421	0.167	0.214	2.523 *
Self-efficacy for CPR					0.787	0.143	0.495	5.513 ***	0.577	0.140	0.363	4.128 ***
Explanatory power	R^2^ = 0.217, aR^2^ = 0.194				R^2^ = 0.332, aR^2^ = 0.318				R^2^ = 0.473, aR^2^ = 0.445			

CPR (cardiopulmonary resuscitation), aR^2^ (adjusted R^2^), * (*p* < 0.05), ** (*p* < 0.01), *** (*p* < 0.001).

**Table 5 healthcare-11-02047-t005:** Hierarchical regression of influencing factors on confidence in performing CPR after CPR education.

Variables	Model 1				Model 2				Model 3			
	B	SE	β	*t*	B	SE	β	*t*	B	SE	β	*t*
Constant	3.208	0.447		7.176 ***	0.672	0.566		1.187	0.897	0.670		1.339
Gender	0.052	0.165	0.027	0.317					0.052	0.157	0.026	0.332
School Grade	0.203	0.078	0.235	2.600 *					0.185	0.076	0.213	2.426 *
Number of CPR education	0.185	0.093	0.178	1.979					0.161	0.088	0.146	1.714
CPR knowledge					0.000	0.060	0.000	0.001	0.016	0.060	0.022	0.269
Attitude toward CPR					0.293	0.152	0.178	1.921	0.317	0.152	0.192	2.080 *
Self-efficacy for CPR					0.380	0.136	0.253	2.795 **	0.324	0.137	0.215	2.363 *
Explanatory power	R^2^ = 0.056, aR^2^ = 0.035				R^2^ = 0.138, aR^2^ = 0.119				R^2^ = 0.180, aR^2^ = 0.143			

CPR (cardiopulmonary resuscitation), aR^2^ (adjusted R^2^), * (*p* < 0.05), ** (*p* < 0.01), *** (*p* < 0.001).

## Data Availability

Data is unavailable due to privacy or ethical restrictions.

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
