# Peer review of "The Effect of Cardiopulmonary Resuscitation (CPR) Education on the CPR Knowledge, Attitudes, Self-Efficacy, and Confidence in Performing CPR among Elementary School Students in Korea"

_healthcare, 2023, doi:10.3390/healthcare11142047_

Round 1

Reviewer 1 Report

Students' training in CPR is an intriguing area of study. One potential limitation of this study is the relatively short interval of two hours between the CPR education and the second testing procedure. This timeframe may make the study resemble more of a training program rather than a comprehensive scientific investigation of the long-term effects of CPR education. It may be worth addressing this point in the discussion section, acknowledging the need for future research with longer follow-up periods. Additionally, including a mention of a planned follow-up study in the conclusions could further highlight the importance of continued research in this field.

Author Response

Students' training in CPR is an intriguing area of study. One potential limitation of this study is the relatively short interval of two hours between the CPR education and the second testing procedure. This timeframe may make the study resemble more of a training program rather than a comprehensive scientific investigation of the long-term effects of CPR education. It may be worth addressing this point in the discussion section, acknowledging the need for future research with longer follow-up periods. Additionally, including a mention of a planned follow-up study in the conclusions could further highlight the importance of continued research in this field.

èThank you very much for your comments. As you mentioned, I added the limitation about present timeframe in the discussion section, and included a mention of a planned follow-up study in the conclusions

Reviewer 2 Report

Thank you for the interesting paper. I have some comments.

1. Ethical issue:

the authors wrote, "Informed consent was received from each participant before  participating in the study." Due to the fact that the research was done among children, I want to make sure that the parents were informed about the research and that they also gave their consent.

2. Methods:

* The authors did not specify where the questionnaire of confidence in performing CPR was taken from.

* How was the suitability of all the questionnaires for the young children tested? Was a pilot carried out?

* Statistical analyses- it should be noted whether the authors did a t-test for paired samples or independent samples. If ×›or paired samples - how were they paired? Were the questionnaires not anonymous? It should be noted. If for independent samples - this is a serious limitation of the study, and this should be noted in the limitations section. It is possible that different children answered the questionnaires before and after the training.

3. Results

* All distributions on all questions (before and after the training) should be presented as an appendix.

* Table 2- t values must be added.

* Table 3- Since these are dichotomous variables (yes/no), chi-square tests should have been done, and this should also be noted in section 2.7. In addition, add the values of the chi-square to the table.

* Table 4- It is more correct to do a hierarchical regression. In the first step, enter only demographic variables. In the second step, the research variables. In the final model, the significant variables are in the previous steps.

4. I have no comments for the discussion section.

Author Response

  1. Ethical issue:

the authors wrote, "Informed consent was received from each participant before  participating in the study." Due to the fact that the research was done among children, I want to make sure that the parents were informed about the research and that they also gave their consent.

èThank you for your comments. We got the informed consents from all children and their parents. A month before the start of the study, we visited the school to explain the study, distributed consent forms for them and their parents to students who showed interested in this study, and conducted the study only for students who submitted two consent forms.

  1. Methods:

* The authors did not specify where the questionnaire of confidence in performing CPR was taken from.

è Thank you for your comments. We edited like that. Confidence referred to confidence in one's ability to perform CPR, and the questionnaires was developed by Park [24].

* How was the suitability of all the questionnaires for the young children tested? Was a pilot carried out?

è Thank you for your comments. We used the questionnaires utilized previous studies conducted on the young children.

* Statistical analyses- it should be noted whether the authors did a t-test for paired samples or independent samples. If ×›or paired samples - how were they paired? Were the questionnaires not anonymous? It should be noted. If for independent samples - this is a serious limitation of the study, and this should be noted in the limitations section. It is possible that different children answered the questionnaires before and after the training.

è Thank you for your comments. We noted that independent T-test was used. As the questionnaires were anonymous, we couldn’t use the paired T-test. As you mentioned, independent T-test cannot measure each student’s change, and that is a limitation of this study. Thus, we tried to prove the effectiveness of CPR education by assessing the mean differences of each variable before and after CPR education. As you commended, we added it in the limitations section.

  1. Results

* All distributions on all questions (before and after the training) should be presented as an appendix.

è Thank you for your comments. We presented all distributions on all questions (before and after the training) as an appendix.

* Table 2- t values must be added.

è Thank you for your comments. We added t values in Table 2.

* Table 3- Since these are dichotomous variables (yes/no), chi-square tests should have been done, and this should also be noted in section 2.7. In addition, add the values of the chi-square to the table.

è Thank you for your comments. We noted that chi-square tests was used, since (yes/no) were dichotomous variables. And, we added the values of the chi-square to the table.

* Table 4- It is more correct to do a hierarchical regression. In the first step, enter only demographic variables. In the second step, the research variables. In the final model, the significant variables are in the previous steps.

è Thank you for your comments. As you recommended, we re-analyzed using hierarchical regression. And, we edited Table 4 with the results of hierarchical regression.

  1. I have no comments for the discussion section.

è Thank you for your comments.

Reviewer 3 Report

Based on the evidence that links CPR started early with a better prognosis, which implies the need to generalize the training of laypeople to recognize situations of cardiorespiratory arrest, the authors study the impact of CPR training on elementary school students in Korea before and after study using questionnaires. The questionnaires focus on; knowledge, confidence, attitude, and self-efficacy. Descriptive statistics, T-tests, and multiple regression were applied. The conclusions are that knowledge increases after teaching but that it is not the principal factor associated with confidence. This led to the conclusion that the important thing will also be to increase in confidence.

The work is well written, easy to read, has adequate methodologies, and aligned discussion and results. Reflection on the limitations is realistic, and the suggested perspectives are reasonable, particularly the need to develop studies on CPR skills. The Korean elemental education system exposes students to multiple CPR teaching actions, which correlates positively. In other educational systems, authorization from parents/guardians may be required in addition to the informed consent of those directly involved. The study reinforces the idea of teaching CPR to elementary school students and the interest in periodically repeating the teaching. 

Some information should be included regarding the way the 120 participants were selected. Probably the absence of a control group was related to the mandatory CPR mandatory proccess

Author Response

Based on the evidence that links CPR started early with a better prognosis, which implies the need to generalize the training of laypeople to recognize situations of cardiorespiratory arrest, the authors study the impact of CPR training on elementary school students in Korea before and after study using questionnaires. The questionnaires focus on; knowledge, confidence, attitude, and self-efficacy. Descriptive statistics, T-tests, and multiple regression were applied. The conclusions are that knowledge increases after teaching but that it is not the principal factor associated with confidence. This led to the conclusion that the important thing will also be to increase in confidence.

The work is well written, easy to read, has adequate methodologies, and aligned discussion and results. Reflection on the limitations is realistic, and the suggested perspectives are reasonable, particularly the need to develop studies on CPR skills. The Korean elemental education system exposes students to multiple CPR teaching actions, which correlates positively. In other educational systems, authorization from parents/guardians may be required in addition to the informed consent of those directly involved. The study reinforces the idea of teaching CPR to elementary school students and the interest in periodically repeating the teaching. 

Some information should be included regarding the way the 120 participants were selected. Probably the absence of a control group was related to the mandatory CPR mandatory process.

è Thank you for your comments. As you recommended, we added the following. “A month before the start of the study, we visited the elementary school in rural areas. After obtaining permission from the principal of the element school, authors visited all classrooms in the 3rd to 6th grades, respectively. We explained the purpose and process of this study and distributed two informed consents to students who showed interest in the study and expressed their intention to participate in the study. Students who completed the submission of informed consents for students and their parents became the final subjects of this study. Informed consents were received from all participants and their parents before participating in the study.”

Reviewer 4 Report

The article submitted for review is related to the very important phenomenon of education related to cardiopulmonary resuscitation (CPR). This issue is part of the broader meaning of the phenomenon of health education in modern societies. Properly conducted health education is one of the most important factors that can counteract risks and help create a healthy society.

The article is properly structured. The introduction sufficiently introduces the subject of the study. The statistical analysis is clear and well described. The results are well described.

I have a few comments:
What was the method of selecting the study participants? (Did they come from urban or rural areas?).
It is important to describe in more detail the tests used. What are the minimum and maximum scores? This is missing for Attitude toward CPR and Self-efficacy for CPR.
Despite the fact that the differences turned out to be statistically significant, the increase in the values of the tested variables was not large - the strength of the effect of the differences does not seem high.

Author Response

The article submitted for review is related to the very important phenomenon of education related to cardiopulmonary resuscitation (CPR). This issue is part of the broader meaning of the phenomenon of health education in modern societies. Properly conducted health education is one of the most important factors that can counteract risks and help create a healthy society.

The article is properly structured. The introduction sufficiently introduces the subject of the study. The statistical analysis is clear and well described. The results are well described.

I have a few comments:
What was the method of selecting the study participants? (Did they come from urban or rural areas?).

è Thank you for your comments. The study participants came from rural areas. As you recommended, we added selecting method of the study participants like following.

“A month before the start of the study, we visited the elementary school in rural areas. After obtaining permission from the principal of the element school, authors visited all classrooms in the 3rd to 6th grades, respectively. We explained the purpose and process of this study and distributed two informed consents to students who showed interest in the study and expressed their intention to participate in the study. Students who completed the submission of informed consents for students and their parents became the final subjects of this study. Informed consents were received from all participants and their parents before participating in the study.”

It is important to describe in more detail the tests used. What are the minimum and maximum scores?

è Thank you for your comments. We added the minimum and maximum scores.

This is missing for Attitude toward CPR and Self-efficacy for CPR.

è Thank you for your comments. We presented all distributions on all questions (before and after the training) as an appendix.

Despite the fact that the differences turned out to be statistically significant, the increase in the values of the tested variables was not large - the strength of the effect of the differences does not seem high.

è Thank you for your comments. As you mentioned, the increase in the values of knowledge, attitude, and self-efficacy after CPR education was significant, but not large. However, the confidence was significantly increased, and the strength of the effect of the differences could be high enough. We expect the increased confidence to raise the bystander CPR, especially, considering the result of question; when you find a cardiac arrest patient in front of you, are you confident of performing CPR? from 51(36.4%) to 112(80.0%) on the number of schoolchildren who marked Yes. Therefore, the result of this study might be meaningful.

Reviewer 5 Report

Title: The effect of cardiopulmonary resuscitation (CPR) education on the CPR knowledge, attitudes, self-efficacy and confidence in performing CPR among elementary school students in Korea.

Reviewer Comments: 

Authors conducted this study to verify the effect of CPR education on knowledge, attitudes, self-efficacy and confidence in performing CPR in Korean elementary school students.  Data was collected before and after CPR education. There was a significant improvement in CPR knowledge, attitudes, self-efficacy and confidence in performing CPR after CPR education. Authors concluded that early CPR education for school students must be crucial and should be conducted regularly and repeatedly.

1.    One of the weaknesses of the study as authors also pointed out is that this study tests only knowledge, but not the real time performance. 

2.    In table 2, comparison of CPR knowledge, attitude, self-efficacy before and after CPR education, is not that significant. 

3.    It’s very difficult for the children in 3rd-5th grade to understand about CPR. I don’t think children are mature enough at that age. Atleast they should start teaching from 5 th grade. 

Author Response

Reviewer Comments: 

Authors conducted this study to verify the effect of CPR education on knowledge, attitudes, self-efficacy and confidence in performing CPR in Korean elementary school students.  Data was collected before and after CPR education. There was a significant improvement in CPR knowledge, attitudes, self-efficacy and confidence in performing CPR after CPR education. Authors concluded that early CPR education for school students must be crucial and should be conducted regularly and repeatedly.

  1. One of the weaknesses of the study as authors also pointed out is that this study tests only knowledge, but not the real time performance. 

è Thank you for your comments. As you mentioned, we acknowledged the limitations of this study. Thus, as we suggested the further study on real time performance, we have just begun the study on CPR skills.

  1. In table 2, comparison of CPR knowledge, attitude, self-efficacy before and after CPR education, is not that significant. 

è Thank you for your comments. As you mentioned, the increase in the values of knowledge, attitude, and self-efficacy after CPR education was significant, but not large. However, the confidence was significantly increased, and the strength of the effect of the differences could be high enough. We expect the increased confidence to raise the bystander CPR, especially, considering the result of question; when you find a cardiac arrest patient in front of you, are you confident of performing CPR? from 51(36.4%) to 112(80.0%) on the number of schoolchildren who marked Yes. Therefore, the result of this study might be meaningful.

  1. It’s very difficult for the children in 3rd-5th grade to understand about CPR. I don’t think children are mature enough at that age. At least they should start teaching from 5 th grade. 

è Thank you for your comments. As you mentioned, third and fourth graders in elementary school may not be at an appropriate age for CPR education. Many studies have tried to find the right time to start CPR education. That is still unclear, but there are many studies showing that CPR education from the lower grades of elementary school is effective. In Korea, CPR education is mandatory from the fifth grade of elementary school, but many schools start CPR education from the lower grades. Therefore, the age of the subject was set from 3rd to 6th grade in elementary school. Despite the inclusion of 3rd and 4th graders in elementary school, the improvement of confidence has been clearly confirmed, so authors stated that CPR education for younger elementary students might be effective. As you recommended, we will investigate the proper starting age of CPR education by assessing the CPR skills before and after CPR education in future study.

Round 2

Reviewer 2 Report

The corrections were made according to the comments. Now the paper can be accepted for publication.

Author Response

The corrections were made according to the comments. Now the paper can be accepted for publication.

==> Thank you so much for comments.